# Why and When Do Employees Hide Their Knowledge?

**DOI:** 10.3390/bs12020056

**Published:** 2022-02-21

**Authors:** Jovi Sulistiawan, Massoud Moslehpour, Fransisca Diana, Pei-Kuan Lin

**Affiliations:** 1Department of Business Administration, Asia Management College, Asia University, Taichung 41354, Taiwan; jovisulistiawan@feb.unair.ac.id (J.S.); writetodrm@gmail.com (M.M.); 2Department of Management, Universitas Airlangga, Surabaya 60115, Indonesia; fransischa.diana@gmail.com; 3Department of Management, California State University, San Bernardino, CA 92407, USA

**Keywords:** distrust, knowledge complexity, knowledge hiding, task performance, task relatedness

## Abstract

This study establishes a theoretical and integrative framework for analyzing the relationship between knowledge hiding and task performance. The existing literature indicates that knowledge hiding is prominent in service sector firms and impedes knowledge transfer. However, the potential determinants and consequences have not been extensively investigated. The objectives of this study are threefold: First, examining the effect of distrust and the complexity of knowledge on knowledge hiding. Second, examining the effect of knowledge hiding on task performance. Third, examining the conditional effect of task relatedness in the relationship between distrust, knowledge complexity, and knowledge hiding. We conducted an online survey by using a Google form to collect our data. We gathered data from 325 salespersons in the business departments of a single firm in Indonesia. To test our hypotheses, we employed Partial Least Square (PLS). The results revealed that distrust and knowledge complexity are critical factors in predicting knowledge hiding. Interestingly, knowledge hiding positively affects task performance. The rationale behind this result is that employees tend to believe that hiding knowledge is a strategy to boost their performance in the short run. The contribution of this study is the suggestion that organizations should implement a knowledge-sharing culture to minimize knowledge hiding.

## 1. Introduction

Knowledge as a resource has gradually become the core resource of organizations in the era of the knowledge economy. The organization tries to stimulate knowledge sharing among its employees, but some prefer hiding their knowledge to maintain their power and position in the workplace. Many studies have been conducted to examine why and when employees share their knowledge, whereas the reasons and outcomes related to knowledge hiding are still limited. Connelly*,* et al. [1] argued that knowledge sharing and knowledge hiding have different motivational drivers. Singh [2] asserted that knowledge, or information, is being kept or hidden by its owners, since it is perceived as a limited or essential resource. In addition, Anand*,* et al. [3] posited that since knowledge is a vital resource and the sharing of knowledge is determined by people who choose who, when, and why to share, specific aspects such as contextual values, boundaries, and a dearth of organizational culture all contribute a significant role in clarifying the underlying reasons why people may not share knowledge, even though the benefits and rewards are apparent. It implies that despite the numerous benefits of knowledge sharing, employees continue to participate in knowledge hiding, which involves concealing information from their coworkers. Several studies have revealed that knowledge hiding has severe consequences because the parties who engage in knowledge hiding and the requesters of the knowledge engage in retaliation and further impair the organization’s ability to acquire an advantage in a highly competitive environment [4,5]. Even though the issue of knowledge hiding has generated researchers’ and practitioners’ interest, the corpus of information around it is still in its infancy and lacks empirical evidence [6].

The act of knowledge hiding has significant consequences for businesses and individual relationships. Knowledge hiding behavior has been connected to negative consequences such as inhibited creativity, increased turnover intention, and extra-role behavior [7,8,9,10]. The issue is that knowledge hiding behaviors are pervasive workplace phenomena that impede knowledge transfer and result in significant financial losses for enterprises [11]. Furthermore, in 2004, the estimated cost of knowledge hiding behavior was USD 31.5 billion per year. The financial impact may have increased in recent years [12]. The available literature indicates that knowledge hiding is prominent in service-sector firms and obstructs knowledge exchange [13]. However, the potential precursors and ramifications of knowledge hiding have received scant attention, and this study seeks to close that gap in the existing literature on knowledge hiding.

This study follows the recommendation from the prior study from Connelly, Zweig, Webster and Trougakos [1] that additional studies be conducted to assess the effect of information hiding on task performance. According to Singh [2], when an individual engages in knowledge hiding from coworkers, he or she will be unable to gain aid or support from colleagues, which may impair individual job performance. Knowledge hiding is an intriguing human action with joyous, nasty, and undesirable outcomes. Wang*,* et al. [14] established an association between knowledge hiding and a decline in individual task performance. However, knowledge hiding behavior is not necessarily considered deviant behavior. A study by Khoreva and Wechtler [15], on the other hand, unearthed that knowledge hiding improves performance. The contradictory findings regarding whether knowledge hiding promotes or diminishes performance urged further investigation.

Knowledge hiding behavior emerges because of distrust, the properties of the knowledge itself, and a response to a social setting [1,16,17]. Therefore, according to Arain*,* et al. [18], it is essential to examine the characteristics of the knowledge itself that can drive employees to hide their knowledge. Furthermore, Issac*,* et al. [19] asserted that employees conceal detailed knowledge because addressing the request for knowledge demands a significant amount of time and effort, hindering employees’ and coworkers’ ability to attain their personal goals. Although numerous studies have been attempted to determine the causes of knowledge concealment, Agarwal*,* et al. [20] highlighted issues about how these determinants exert their consequences and whether their intensities change across circumstances. Furthermore, Jiang*,* et al. [21] recommended that future research elucidate the circumstances behind the determinants of knowledge hiding. Thus, this study proposed task relatedness as a conditional variable that may affect the magnitude of distrust and knowledge complexity on knowledge hiding.

To ensure that knowledge transfer between peers is accomplished smoothly in an organization, the organization must emphasize the interdependence of employees and others through assigned duties. Task relatedness refers to the degree to which one’s work is dependent on the work of others or the degree to which peers’ jobs are interrelated [22]. It has been demonstrated that task relatedness mitigates the unfavorable conduct of individual knowledge hiding from others. The odds of employees concealing knowledge from their coworkers will decrease if the knowledge or information sought by others is directly or mainly relevant to the tasks allocated to them [1]. As such, the objectives of this study are threefold: First, examining the effect of distrust and the complexity of knowledge towards knowledge hiding behavior. Second, examining the effect of knowledge hiding on task performance. Third, examining the conditional effect of task relatedness in the relationship between distrust, knowledge complexity, and knowledge hiding. To achieve our objectives, this study employs the PLS-SEM method in multiple ways. First, to obtain the validity and reliability of all measurements, this study employs validity and reliability analysis. Second, after the measurements are validated, the structural path analysis is used to test our proposed hypotheses.

Even though the direct effect of distrust and knowledge complexity on knowledge hiding have been conducted in prior studies, the contextual factor which amplifies such a relationship is still limited. Therefore, this study provides several theoretical and practical contributions related to knowledge hiding behavior. First, this study will provide a broader context for knowledge hiding by investigating the causes and effects of knowledge hiding behavior. Second, this study will assist managers in comprehending how task relatedness acts as a situational factor in the association between distrust and knowledge complexity in knowledge hiding. Finally, this study broadens the scope of knowledge-hiding studies by studying notions created in a developing country such as Indonesia.

The remainder of this study is as follows, second 2 analyzes prior research on knowledge hiding behavior and the offered hypotheses, and Section 3 provides a complete overview of the methodologies employed in this study. Section 4 contains the findings; Section 5 discusses the practical and theoretical ramifications. Lastly, Section 6 summarizes the findings, discusses the limitations, and makes recommendations for future studies.

## 2. Literature Review and Hypothesis Development

### 2.1. Knowledge Hiding (KH)

Knowledge hiding (KH) has gained scholars’ attention, primarily related to human resource management and organizational behavior. The study of KH is mainly developed from the concept of knowledge management [23]. Early research into KH was conducted by Campbell*,* et al. [24]. Their study, related to data withholding by scientists, revealed that scientists were most likely to be the victims of KH from other scientists who withheld their research results. Later, in 2008, Webster et al. [25] separated KH and knowledge withholding. According to their study, KH is considered as one type of knowledge withholding.

Further, the study from Ford and Staples [25] made a critical contribution in explaining the difference between KH and knowledge sharing. Knowledge sharing and KH are not in the same continuum. It implies that the lack of knowledge sharing is different from KH. Lack of knowledge sharing occurs mainly because the employees simply do not know; they do not attempt to hide the knowledge. In 2012, Connelly et al. expanded the research related to KH. Since then, many studies have used the study from Connelly et al. as the foundation of the KH research.

KH refers to a person’s deliberate efforts to conceal or withhold knowledge related to tasks, ideas, and the information requested by others, where such behavior has the potential to harm working relationships and foster distrust between individuals, create knowledge gaps, and reduce individual performance, all of which harm organizational performance [1,7]. When an employee refuses to provide knowledge or information, this does not always indicate that they have committed an act of deception [1,11,26]. In some instances, KH demonstrates positive goals and outcomes, such as white lies, where the behavior is carried out with the intent of safeguarding others’ feelings, maintaining the confidentiality of knowledge or information, and safeguarding the interests of others, including those of the organization [26]. As a result, concealing knowledge is not always a negative behavior [15,26].

Connelly, Zweig, Webster, and Trougakos [1] distinguished three types of KH behavior: evasive hiding, playing dumb, and rationalized hiding. Employees engage in evasive hiding when they employ evasive techniques to avoid disclosing information or when they share incomplete or even inaccurate information [27]. Evasive hiding entails some lying, as the conduct is characterized by offering inaccurate or incomplete information [1]. Evasive hiding is regarded as a dishonest approach that can elicit punishment from those who witness such behavior [9].

Playing dumb is one method of concealing knowledge [28]. As with evasive hiding, playing dumb is a deceptive approach that can elicit retaliation from others who witness the activity [8,11,29]. Someone employs this technique by claiming ignorance of the knowledge sought by others and refusing requests to impart this knowledge. Employees that use the tactic of playing dumb seek to avoid sharing their knowledge with others [26,27]. In other words, employees pretend not to understand the questions posed by others, prefer to defer responses, and disregard pleas for knowledge when others need assistance [30].

Rationalized hiding occasionally entails deception [1,31]. Someone who uses reasoned hiding is more intelligent than someone who employs evasive hiding as a KH approach. Individuals can rationalize KH by justification or by blaming others. Individuals excuse their hiding by claiming that they cannot give the requested information because it is confidential or pertains to corporate policies [27]. Engaging in rationalized hiding, individuals frequently respond diplomatically to knowledge requests, stating that the requested knowledge is confidential, and the individual is unable to share it, or that the individual is not the best person to ask, forcing colleagues to consult with others [32].

### 2.2. Distrust (DIS) and Knowledge Hiding (KH)

Yuan*,* et al. [33] define distrust (DIS) as a subjective lack of confidence in others during interpersonal interactions, based on the belief that others are unpleasant and would hurt people by exploiting their vulnerabilities. The term “social interaction” is synonymous with “social exchange.” Individuals can refuse to transfer knowledge as a way to preserve a competitive edge in social contacts with other employees in an organization. Further, Blau [34] argued that interpersonal distrust is the foundation for unproductive interpersonal relationships. Thus, Connelly and Zweig [11] addressed that in the era of knowledge economy, the distinctive value of knowledge causes individuals to conceal their information from requesters to avoid losing their knowledge superiority, exploiting others when a “knowledge inquiry” occurs. This underlying unwillingness of “knowledge support” is generally viewed negatively by requestees, eroding the employees’ current pleasant emotional foundation; as a result, interpersonal distrust develops, and, quite significantly, the propensity of employees to socialize is harmed, and interpersonal distrust in relationships emerges.

Between individuals, information concealing occurs when the strength of their connection affects how individuals respond to demands for knowledge from other individuals [1]. Essentially, individuals frequently participate in conduct that benefits others voluntarily and spontaneously [34]. According to Connelly and Zweig [11], the nature of social trade began to develop and increase in lockstep with the times. It occurs because individuals eliminate existing duties and create new ones to establish trust with various parties who show interest in them. According to Blau’s social exchange theory [34], the reciprocal history between persons might motivate them to be involved in KH. Previous studies revealed that employees tend to display deviant behaviors when there is a perception of DIS [16,35,36]. Based on that rationale, KH is a response to DIS, the properties of knowledge, and as a response to the social situation [5]. In line with the social exchange theory, employees tend to have higher motivation to balance the unfavorable social exchange. Specifically, as the DIS occurs toward the peer, the employee is most likely to balance the negative social exchange by engaging in KH behavior. Thus,

**Hypothesis** **1** **(H1).** 
*DIS has a positive effect on KH.*


### 2.3. Knowledge Complexity (KC) and Knowledge Hiding (KH)

The core relationship among employees may affect their behavior, especially in the management of knowledge in the workplace. However, it is also essential to consider the properties of the knowledge itself that may affect the employees’ intentions to hide it [1]. One of the characteristics of knowledge is complexity [11,37]. Therefore, the complexity of the knowledge affects the knowledge transfer of the employees. The rationale behind this behavior is that employees devote the time and effort to acquire complex knowledge for their career advancement and performance, intending to enhance their competencies. Therefore, when it comes to a request from peers for incredibly complex knowledge, it will be difficult for the requester to acquire the knowledge because the “owner” of the knowledge tends to avoid losing this complex knowledge, thereby strengthening a peer’s competitive edge. It is analogous with the psychological ownership theory which explains that employees tend to establish a sense of ownership as they devote their time, energy, efforts, and money to acquiring specific knowledge [23]. Further, once they develop strong bonds with their knowledge, they will be less likely to share their knowledge because they perceive it as a pitfall to their ownership.

Several studies have been conducted to examine the effect of KC on KH behavior. For instance, research from Peng*,* et al. [38] focused on the complexity of knowledge, which is also acknowledged as a significant component in defining employee hiding practices. Connelly and Zweig [11] revealed that individuals frequently conceal knowledge, especially sophisticated knowledge. Kumar Jha and Varkkey [27] asserted that when the demand for knowledge is straightforward, individuals are likely to utilize playing dumb or reasoned hiding tactics to conceal knowledge because individuals believe that pretended ignorance of the answer or using diplomatic responses are the quickest ways to halt requests for knowledge from others. On the other hand, employees frequently employ evasive hiding tactics to conceal knowledge when knowledge queries are difficult and complex because they believe that by dodging or providing incomplete answers, they have aided others, even if the benefit is insignificant.

**Hypothesis** **2** **(H2).** 
*KC has a positive effect on KH.*


### 2.4. Knowledge Hiding (KH) and Task Performance (TP)

Singh [2] conceptualized employee task performance (TP) in two distinct areas. First, the organization requires that the activities contribute to the technical core and are officially acknowledged as part of the job. Second, the tasks must contribute to the technical core, a critical quality that enhances the performance components. In addition, TP is described as activities stated and required by an employee’s roles and consequently, demanded, evaluated, and compensated by the organization [15].

This study followed the suggestion of Connelly, Zweig, Webster, and Trougakos [1] to further examine the relationship between KH and individual TP. KH will enhance employees’ performance in the short-term period. The agency theory provides the rationale for such a relationship. This study draws from the agency theory and assumes that information transfer and assimilation are intended to maximize organizational value. This knowledge also becomes essential and crucial to individuals, resulting in improved short-term job performance [15,39].

Further, since knowledge becomes an indispensable resource in an organization, employees who engage in KH behavior increase their peers’ dependence, thus increasing the bargaining power in the organization. By concealing exclusive knowledge, employees potentially expand their knowledge bases and generate unequal value for the firm over time. As a result of owning and controlling knowledge and participating in knowledge concealment, employees may enhance their TP.

**Hypothesis** **3** **(H3).** 
*KH has a positive effect on the TP.*


### 2.5. The Moderating Role of Task Relatedness (TR)

The employees’ tendency to engage in KH is subject to the characteristics of the task [19]. One of the task characteristics that may hinder employees from hiding their knowledge is task relatedness (TR). Peng, Wang, and Chen [38] defined TR as the extent to which colleagues rely on each other to accomplish and fulfill their assigned tasks effectively. Hernaus, Cerne, Connelly, Vokic, and Škerlavaj [5] argued that TR provides a distinct perspective on job characteristics beyond motivating factors. While empirical studies advocate for the reciprocal design of tasks and incorporate reciprocal methods (such as task relatedness) into their knowledge-sharing previous studies, additional study is needed to determine the consequences of such job-related aspects. When a complex activity necessitates the knowledge obtained through broad cross-functional collaboration, the resulting interactions grow highly complicated. In general, these contacts will foster deeper trust between departments and decrease concealed knowledge activities within the organization. TR has been linked to beneficial team dynamics, including intragroup support, collaboration, and group cohesiveness [38]).

TR is considered a contextual factor that may hinder the negative consequences of interpersonal distrust and KC toward KH behavior. Extreme task-related levels demand collaboration among peers to obtain critical resources, such as knowledge [5,40]. A prior study by Černe, Nerstad, Dysvik, and Škerlavaj [8] revealed that in order to perform the assigned tasks, interrelated employees are less likely to conceal knowledge from peers, even when they lack trust in them. TR does not provide employees with many options; instead, job factors hinder employees’ chances to engage competitively, regardless of how distrustful they are of one another. The same rationale is applied to the relationship between KC and KH. As the employees engage in a high interrelatedness, the members realize that they are free to communicate, share information, and contribute to accomplishing the task regardless of how complex the knowledge is. Thus,

**Hypothesis** **4a** **(H4a).** 
*The association between DIS and KH is moderated by TR. When TR is high, the relationship between DIS and KH will decrease.*


**Hypothesis** **4b** **(H4b).** 
*The association between KC and KH is moderated by TR. When TR is high, the relationship between KC and KH will decrease.*


## 3. Research Methods

### 3.1. Sample and Data Collection

An online survey was conducted at the beginning of 2021 with a salesperson representing individual branches of Indonesia’s largest financial services firm, whose main service is providing consumer loans with movable collateral. The firm relies heavily on a knowledge management system to achieve higher organizational performance. Furthermore, the performance management system in the organization is based on organizational targets that cascaded to individual and team performance targets thus, the practice of knowledge hiding will affect both an individual’s performance, as well as the organization.

The data collection was completed in two stages. In the first stage, we contacted the company’s vice presidents and sent out the proposed survey. Data were collected from two sources, namely salespersons and their superiors. In 2020, the total number of salespersons was 983. The corporate vice presidents provided us with list of 450 employees. The superiors were involved in this study to measure their subordinates’ performance. In the second stage, we contacted 450 employees and requested them to participate in the study. There were 340 responses, with 325 qualifying as usable data for further analysis. We employed Slovin’s formula to determine the minimum sample [41]. We obtained 284 as minimal respondents with a 5% margin of error, and the number of respondents involved in this study was 325. Therefore, the number of responses exceeded the minimum requirement of sample representativeness. Table 1 shows that our sample consisted of male 50.9% and female 49.1%. Most of our respondents were 21–27 years old (57.3%), tenure was 1–2 years (48.2%), and full-time employment status was 70.9%.

### 3.2. Measures

All measures used a response scale where 1 was “strongly disagree”, and 7 was “strongly agree”. This measurement scale was used in several prior studies. The English version of the survey instrument was translated into Bahasa before it was distributed to the responders.

DIS was measured by Lewicki, et al. [42] using six items. KC was measured by Pérez-Luño, et al. [43] using four items. TR was measured by Pierce, et al. [44] using seven items. For KH, we used three dimensions from KH by Connelly, Zweig, Webster, and Trougakos [1], consisting of four items for each dimension, the dimensions being evasive hiding, rationalized hiding, and playing-dumb. The measurements of these variables are presented in Appendix A. TP was measured by using the key performance indicators from the organization, which consists of four items: the amount of fee-based income, the realization of the number of savings customers, the realization of the number of active users, and the number of one-on-one meetings with the customers. We used the employees’ KPI to make the measurement suitable for the context of our study. We asked the superiors to rate their salespersons’ TP. The rationale behind this method is to reduce the social desirability bias.

## 4. Results

### 4.1. Validity and Reliability

We used three types of validity to evaluate the measurements in order to ensure the quality of our outer model. First, we ensured the convergent validity by assessing the score of the loading factor in each measurement and the average variance extracted (AVE), with 0.50 as the cut-off value [45]. In the first run, we eliminated three indicators (KH8, KH9, KH11) from the KH construct because they did not meet the cut-off value. After we eliminated the invalid indicators, we reran the data, which is presented in Table 2. Table 2 and Figure 1 show that the indicators’ loadings exceeded the acceptable level of 0.50. Moreover, the AVE scores of all the variables were above 0.50, providing strong support for convergent validity. Second, we ensured the reliability of our measurements by checking the composite reliability score for each variable. The variables are considered reliable if the composite reliability score is higher than 0.70 [45]. As shown in Table 2, the composite reliability (CR) scores for each variable are above the required threshold of 0.70.; thus, the evidence can be used to support the reliability of the construct.

Finally, we assessed the discriminant validity using a recommendation from Henseler, et al. [46], namely the heterotrait-monotrait correlations ratio (HTMT) criterion and the Fornell–Larcker criterion. As shown in Table 3, all HTMT values range between 0.095 and 0.754, therefore not exceeding the threshold for HTMT of 0.85. As seen in Table 4, the AVE scores for each variable are greater than the correlation with all other variables. Based on all of the assessments, it is determined that all of the measurement models show convergent and discriminant validity.

### 4.2. Hypothesis Testing

#### 4.2.1. Direct Effect

To test our proposed hypotheses, we evaluated the path coefficients. We used the recommendation from Hui-Wen Chuah, et al. [47] by using the bootstrapping method with 5000 subsamples in order to determine the significance of the proposed hypotheses. Figure 2 and Table 5 show that the direct effect of DIS is positively and significantly affecting KH (β = 0.438, *p* < 0.001); thus, H1 was supported. Hypothesis 2 predicts that there is a positive relationship between KC and KH. In Figure 2, KC has a positive effect on KH (β = 0.316, *p* < 0.001), thus supporting H2. Furthermore, KH has a strong positive relationship to TP (β = 0.961, *p* < 0.001); thus, H3 was supported

#### 4.2.2. Moderating Effects

Concerning the moderating effects, we employed the two-stage procedure to determine whether the relationship between DIS and KC on KH are moderated by TR. The advantage of the two-stage procedure is that it can be used for exogenous or moderating variables for both reflective and formative indicators. Additionally, this strategy is superior to other methods (e.g., product indicator and orthogonalizing) in terms of statistical power, single effect estimation, and parameter recovery [47]. Figure 1 and Table 4 show the relationship between DIS and KH in relation to TR. Hypothesis 4 proposed that TR diminishes the positive effect of DIS on KH. In contrast to our proposed hypothesis, our results showed that TR strengthens the positive effect of DIS on KH (β = 0.170, *p* < 0.01); thus, hypothesis 4a was not supported. Hypothesis 4b predicts that TR weakens the relationship between KC and KH; the result supports hypothesis 4b (β = −0.243, *p* < 0.001). We employed simple slope analysis to further visualize the interaction effect. Figure 3 and Figure 4 show the interaction effect between DIS, KC, TR, and KH. The relationship between task complexity and KH is diminished when the task relatedness is high. It implies that a high level of TR will reduce the KH, especially when the complexity of the knowledge is high, thus supporting hypothesis 4b.

## 5. Discussions

This study established a theoretical and integrative framework for analyzing the relationship between knowledge hiding and TP, as well as the antecedents of KH. Specifically, this study identified why employees tend to engage in KH behavior. This study sheds light on the contextual function of TR, which might moderate the relationship between DIS, KC, and KH.

The findings of this study corroborate earlier studies, indicating that employees lack of trust in their coworkers increases based on the higher the employee’s intention is to conceal knowledge from his or her coworkers. The results of this study support previous studies showing that DIS frequently motivates employees to hide information from coworkers [1,16,35]. According to the notion of social exchange, employees are more likely to avoid a negative social exchange. Employees are more prone to engage in knowledge concealment behavior to counteract an unfavorable social interaction when they develop a DIS for their coworkers. Further, the strong positive association between DIS and KH is due to the employees’ tenure. In the early years of a worker’s employment with an organization, difficulties arise as she/he is confronted by the expectations of the organization and of her/his superiors and subordinates, if any [19]. In these instances, a lack of trust may contribute to knowledge concealment tendencies.

The complexity of knowledge is one of the variables that determine why employees hide information Connelly and Zweig [11] asserted that employees frequently conceal sophisticated knowledge, often because the nature of the knowledge request demands additional time and effort, hence reducing the employee time available to complete their own tasks. Employees invest time and effort in gaining sophisticated knowledge to improve their careers and perform capably, as well as to upgrade their skills. When colleagues request knowledge, particularly complicated knowledge, it will be problematic for the requester to gain the knowledge since the “holder” of the knowledge has a desire to avoid losing complex knowledge. The result of this study supports the previous study showing that the characteristics of knowledge, namely KC, becomes a strong predictor for employees’ KH behavior [33]. It will be highly unlikely to share the knowledge if it is complex or sophisticated, because hiding such knowledge is considered as a defense mechanism to avoid the loss of key resources.

This study reveals that KH has a positive effect on employees’ TP. The relationship between these two constructs remains inconsistent. Thus, this study supports previous works indicating that there is a positive outcome from a negative behavior [14]. Furthermore, because knowledge has become a critical resource in business, individuals who participate in knowledge concealment behaviors boost their peers’ reliance on them, hence increasing their bargaining power [15]. Employees can potentially increase their knowledge bases and provide uneven value for the organization over time by concealing unique knowledge. The finding of this study, that knowledge concealment is positively associated with job performance, is consistent with the agency theory, which postulates that individuals may generate self-interested strategies to mislead their peers in order to achieve benefits such as improved job performance. As a result, when employees engage in KH, they easily generate justifications for their behavior in terms of greater inventive performance. Thus, it may enhance their work performance because of their ownership and control of knowledge, as well as motivate their participation in KH.

The interesting finding in this study is how TR elevates the association between DIS and KH. The result shows that a high level of TR will enhance the propensity of employees to engage in KH behavior, especially when the level of DIS is high. It implies that TR amplifies the effect of DIS on KH. When employees lack trust in their peers and task interrelatedness is high, employees believe that sharing knowledge implies supporting colleagues in accomplishing performance targets, which will hurt the proprietor. Moreover, the greater the task-interrelatedness, the more prone it is to social loafing, which is why employees even more strongly attempt to conceal their knowledge due to the possibility of social loafing [38]. On the other hand, TR diminishes the negative effect of KC on KH. This implies that the more complex the knowledge, the less likely the employee will be to hide it, especially when the tasks are interrelated.

### Practical Implications

The findings of this study offer several implications for managers. First, the findings suggest that the properties of knowledge will affect employees’ decisions about whether to distribute or conceal knowledge. The more complex the knowledge, the most likely the employees will be to hide it from their colleagues. It implies that employees are likely to perceive that simple knowledge is easy to share, but tacit knowledge becomes their own; thus, they keep it as a critical resource. This finding suggests managers should identify the impact of knowledge properties when persuading employees to participate in knowledge-sharing activities. Additionally, managers need to value sophisticated and deep knowledge and consider the features of knowledge prior to inviting employees to take part in knowledge-sharing activities. Managers may use soft management strategies to demonstrate their compassion and to overcome employees*’* reluctance to share in the case of complex knowledge. Second, this study reveals that interpersonal relationships are an important factor when deciding to share or hide knowledge. When employees do not trust their peers, it is most likely that KH will occur. This result suggests that managers recognize that trust becomes the foundation of the organization. To motivate the employees to actively engage in knowledge-sharing practices, the level of trust among them should be increased. Managers might strive to boost employees*’* impressions of their coworkers*’* truthfulness by stressing a collective identity or by emphasizing examples of demonstrated trustworthiness.

Third, our interesting finding is KH enhances TP. This implies that KH behavior is not necessarily harmful or dysfunctional because those who participate in KH behavior attempt to protect their own, or even their organizations’, interests. Employees are motivated to enhance their performance; nevertheless, they frequently guard own egos and respond to the social pressure to outperform their peers. Thus, employees violate ethical standards, opting to hide knowledge as a highly effective short-term tactic for improving their performance. Managers should foster an ethical work environment that emphasizes employee trust and knowledge sharing. The establishment of a common identity and the development of a positive corporate culture may all contribute to dissuading employees from concealing knowledge. Lastly, our study reveals that task-interrelatedness does not necessarily hinder KH behavior. A significant level of DIS, combined with a high level of task interrelatedness, will encourage employees to hide their knowledge. It is vital that managers address this issue, since sales tasks are interrelated among employees, and the organization’s success depends on reducing the level of DIS among the employees.

## 6. Conclusions and Limitations

### 6.1. Conclusions

This study investigated the cause and effect of KH. We proposed that KC and DIS are the main reasons why employees engage in hiding knowledge. Additionally, we also examined the conditional effect of task interrelatedness in the relationship between KC and DIS and its effect on KH. The study findings indicate that KH is significantly influenced by the level of DIS. In addition, the level of KC has a negative effect on KH. It implies that the higher the complexity of knowledge, the greater the employees’ tendencies to conceal their knowledge. KH *behavior has a positive effect on both employees and the organization, such as increasing performance.* Regarding the conditional effect of TR, the finding of this study confirms that TR weakens the relationship between KC and KH, whilst strengthening the relationship between DIS and KH. We tested our proposed hypotheses by using PLS-SEM, and also evaluated the validity and reliability of our measurements. Our results reflect the study’s significance for both theoretical and practical KH behavior. There are several ways in which this research contributes to the KH research domain. First, this study provides theoretical contributions by assessing the cause and effect of KH behavior, thus offering a way for industry, especially in Indonesia, to effectively minimize KH behavior. Second, this study indicates that KH is not necessarily bad behavior, since it boosts employees’ performance in the short run. Third, this study identifies TR as a conditional effect that influences the impact of DIS and KC on KH, whilst many prior studies mainly considered TR as an antecedent of KH behavior.

### 6.2. Limitations and Future Research

The results of this study identified the antecedents and the consequences of KH behavior. Several limitations, as well as future research suggestions, apply to this study. First, the attributes in this study were derived from a previous study which may not cover the holistic attributes of KH behavior. The current research does not examine DIS, KC, and TR for each of the dimensions of KH, such as evasive hiding, playing dumb, and rationalized hiding, or correlate each dimension of KH with TP variables. Thus, future research is expected to examine each variable associated with each level of knowledge concealing, namely evasive hiding, playing dumb, and rationalized hiding. Second, this study is limited to only a single company. To increase generalizability, future studies may include a variety of businesses. Third, this study used cross-sectional data whereas, in order to capture the loop of DIS, as well as to compare the effect of KH to short- and long-term performance, longitudinal data or research should be conducted in the future. Fourth, the current study was conducted using the full quantitative method, whilst KH is commonly perceived as a negative behavior. Thus, future studies may conduct the research by employing both qualitative and quantitative methods.

## Figures and Tables

**Figure 1 behavsci-12-00056-f001:**
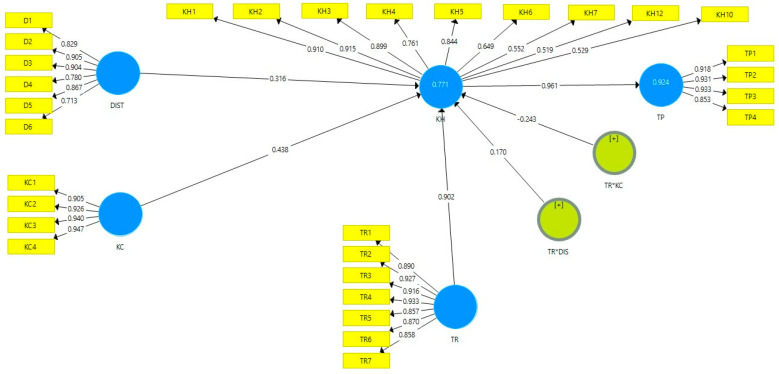
PLS Structural Path.

**Figure 2 behavsci-12-00056-f002:**
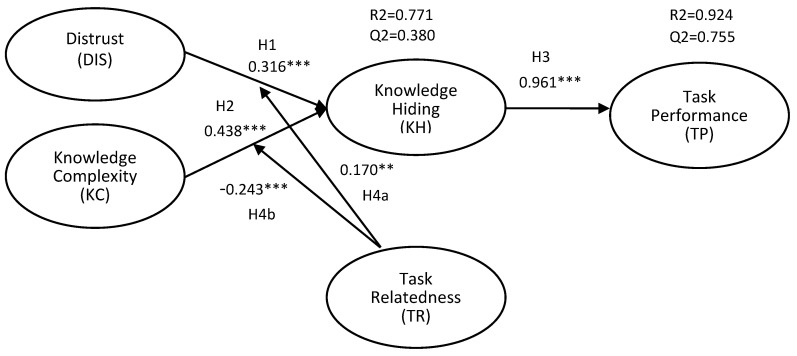
The results for the structural path. Notes: ** *p* < 0.01, *** *p* < 0.001.

**Figure 3 behavsci-12-00056-f003:**
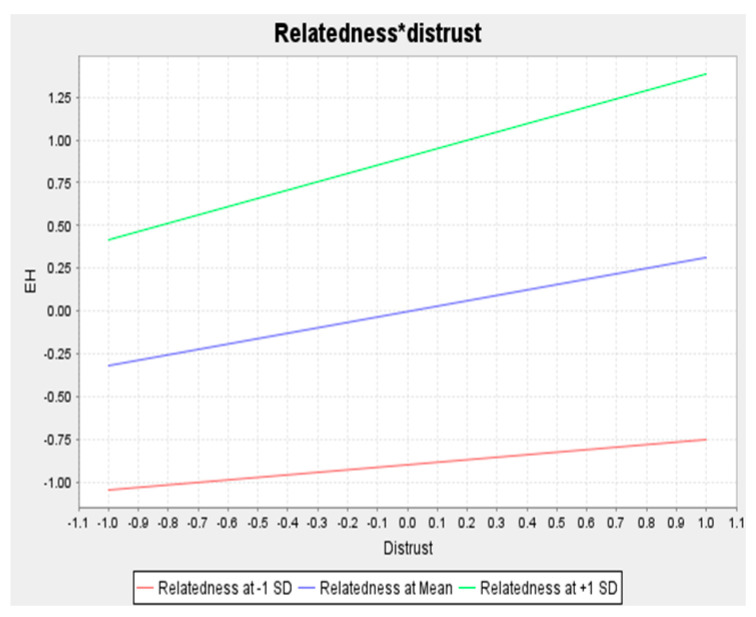
The moderating effect of task relatedness on complexity and knowledge hiding.

**Figure 4 behavsci-12-00056-f004:**
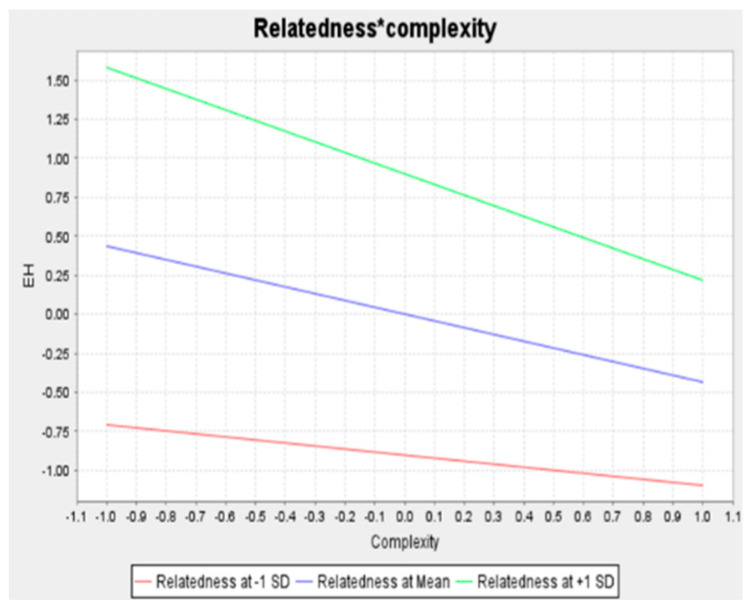
The moderating effect of task relatedness on distrust and knowledge hiding.

**Table 1 behavsci-12-00056-t001:** Characteristics of the respondents.

Demographic	Classifications	Frequency	Percentage
Gender	Male	165	50.8%
	Female	160	49.2%
Age	18–27 years old	186	57.2%
	28–34 years old	98	30.2%
	35–41 years old	27	8.3%
	Above 41 years old	15	4.6%
Employment status	Full-time	230	70.8%
	Part-time	95	29.2%
Tenure	Less than 1 year	74	22.8%
	1–2 years	157	48.3%
	2–4 years	80	24.6%
	Above 4 years	15	4.6%

**Table 2 behavsci-12-00056-t002:** The validity and reliability results.

Variable	Indicators	Loading Factor	*t*	AVE	α	CR
Knowledge Hiding (KH)	KH1	0.910	62.480	0.628	0.888	0.903
	KH2	0.915	65.943			
	KH3	0.899	44.052			
	KH4	0.761	14.201			
	KH5	0.844	32.667			
	KH6	0.649	10.259			
	KH7	0.552	5.707			
	KH12	0.519	4.822			
	KH10	0.529	5.352			
Distrust (DIS)	D1	0.829	25.040	0.699	0.913	0.933
	D2	0.905	71.207			
	D3	0.904	77.275			
	D4	0.780	25.122			
	D5	0.867	34.177			
	D6	0.713	20.983			
Knowledge Complexity (KC)	KC1	0.905	17.156	0.864	0.947	0.962
	KC2	0.926	54.372			
	KC3	0.940	54.070			
	KC4	0.947	109.292			
Task Performance (TP)	TP1	0.918	79.892	0.827	0.930	0.950
	TP2	0.931	94.290			
	TP3	0.933	79.884			
	TP4	0.853	25.392			
Task Relatedness (TR)	TR1	0.890	54.671	0.798	0.958	0.965
	TR2	0.927	77.082			
	TR3	0.916	72.813			
	TR4	0.933	100.938			
	TR5	0.857	35.450			
	TR6	0.870	31.608			
	TR7	0.858	40.673			

**Table 3 behavsci-12-00056-t003:** The Heterotrait-Monotrait Ratio (HTMT).

	KH	DIS	KC	TP	TR
1. KH	-				
2. DIS	0.392	-			
3. KC	0.144	0.455	-		
4. TP	0.702	0.095	0.238	-	
5. TR	0.323	0.321	0.754	0.440	-

Notes: KH—knowledge hiding; DIS—distrust; KC:—knowledge complexity; TP—task performance; TR—task relatedness.

**Table 4 behavsci-12-00056-t004:** The Fornell–Larcker Criterion.

	KH	DIS	KC	TP	TR
1. KH	**0.792**				
2. DIS	0.422	**0.836**			
3. KC	0.231	0.555	**0.923**		
4. TP	0.762	0.196	0.315	**0.909**	
5. TR	0.439	0.497	0.824	0.550	**0.893**

Notes: KH—knowledge hiding; DIS—distrust; KC—knowledge complexity; TP—task performance; TR—task relatedness. The diagonal values in bold are the square root of AVE.

**Table 5 behavsci-12-00056-t005:** The results of hypothesis testing.

Hypothesis		β	*p*-Value	Bias-Corrected 95% CI		Remarks
				Lower	Upper	
H1	DIS→KH	0.316	0.000	0.216	0.421	Supported
H2	KC→KH	0.438	0.000	0.281	0.582	Supported
H3	KH→TP	0.961	0.000	0.946	0.973	Supported
H4a	TR*DIS→KH	0.170	0.011	0.046	0.282	Not supported
H4b	TR*KC→KH	−0.243	0.001	−0.357	−0.121	Supported

Notes: KH—knowledge hiding; DIS—distrust; KC—knowledge complexity; TP—task performance; TR—task relatedness.

## Data Availability

Not applicable.

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
