# Peer review of "Why and When Do Employees Hide Their Knowledge?"

_behavsci, 2022, doi:10.3390/bs12020056_

Round 1

Reviewer 1 Report

The authors map a relatively problematic area, hiding knowledge, which is currently a significant factor that affects working conditions and the working environment.
Unfortunately, the authors limited the research to only one company, which could have distorted the results. The authors used scientific methods, stated hypotheses and a mathematical statistical apparatus.
It would be appropriate to compare the study with other companies or with other work environments, which, however, the authors stated in the conclusions.

Author Response

Dear reviewer, we greatly appreciate your constructive comments! Please find below our responses to each of your comments.

Point 1: the authors limited the research to only one company, which could have distorted the results.

Response 1: Thank you very much for your comments about this point. We realized that limiting to only one company might distort the result. However, we have wanted to keep the scope of the study in a manageable framework. In addition, we also addressed this issue by putting this in the limitation and suggestion for future study in order to enhance the generalizability.

Point 2: It would be appropriate to compare the study with other companies or with other work environments, which, however, the authors stated in the conclusions.

Response 2: Thank you very much for your comments about this point. Yes, indeed this is our next step in expanding this topic.

Reviewer 2 Report

The study points to an essential and highly topical issue. With growing stocks of knowledge, those organizations that operate knowledge management profitably have an advantage. Few seem able to do this. So the topic is very important. The study has some ambiguities that should be clarified: Re point 4.1: The representation of validity and reliability is not entirely clear from the text. Re point 4.2.2: Here, too, the performance should be more understandable. The wording contains overly complicated sentences that make it difficult for the reader to understand. Re point 5: The discussion is well presented. Re point 6.1 The conclusion is not presented very clearly. Please specify more clearly.

Author Response

Response to Reviewer 2

This is an interesting study,

Thank you very much for your constructive and kind encouragement. Dear reviewer, we have gone through your constructive suggestions and highlighted our changes in the revised manuscript. Thank you in advance for your time and collaboration to make this manuscript publishable with a high standard and quality.

However, we would like to draw attention to several aspects of the current manuscript that might be improved or needs to be clarified.

Comments

Responses

Re point 4.1: The representation of validity and reliability is not entirely clear from the text

Thank you very much for your fruitful comments.

Improved.

We have elaborated the explanation of validity and reliability in section 4.1.

Please refer to lines 316-323, table 2 and figure 1.

Re point 4.2.2: Here, too, the performance should be more understandable. The wording contains overly complicated sentences that make it difficult for the reader to understand

Thank you very much for your fruitful comments.

Improved.

We have elaborated the explanation of section 4.2.2.

Please refer to line 362-364; 368-371; 373-375.

Re point 5: The discussion is well presented.

Thank you very much for your fruitful comments.

Re point 6.1 The conclusion is not presented very clearly. Please specify more clearly.

Thank you very much for your fruitful comments.

Improved.

We have written and elaborated the conclusion by adding several sentences related to the result of our study.

Please refer to lines 475-480

The study findings indicate that knowledge hiding is significantly influenced by the level of distrust. In addition, the level of knowledge complexity has a negative effect on knowledge hiding. It implies that the higher the complexity of knowledge, employees tend to conceal their knowledge. Knowledge hiding behavior has a positive effect on both employees and the organization, such as increasing performance. Regarding the conditional effect of task-relatedness, the finding of this study confirms that task-relatedness weakened the relationship of knowledge complexity and knowledge hiding, whilst strengthening the relationship between distrust and knowledge hiding.

Reviewer 3 Report

The research deals with extremely important and interesting issue of behavioural studies. The paper mets all formal requirements, it is logically presented and well-structured. Hypotheses are in line with the research objectives and conclusions. Just some moments should be considered by Authors in order to improve overall quality of the paper:

(1) describing the measures for own research (lines 303-306), authors refer to studies with appropriate description. It would be better if authors provide the list of measures (or relevant questions from questionnaire) in Appendices for the manuscript - the vizualization of results and their understanding during the reading would be essentially higher; This gap can be easily filled, considering the statement "we had the English version of the instrument...";

(2) the representativeness of the sample of "325 usable data" should be justified by authors using the data about overal number of employees of the chosen kind of activity. Now we have enough information about demography of the sample and methods of data collections. However, it is doubtful that the results are reliable in the light of representativeness requirements. I believe that the evidences of sample representativeness will not take a lot of time considering the advanced method of analysis used in the research.

Author Response

Response to Reviewer 3

This is an interesting study,

Thank you very much for your constructive and kind encouragement. Dear reviewer, we have gone through your constructive suggestions and highlighted our changes in the revised manuscript. Thank you in advance for your time and collaboration to make this manuscript publishable with a high standard and quality.

However, we would like to draw attention to several aspects of the current manuscript that might be improved or needs to be clarified.

Comments

Responses

1) describing the measures for own research (lines 303-306), authors refer to studies with appropriate description. It would be better if authors provide the list of measures (or relevant questions from questionnaire) in Appendices for the manuscript - the vizualization of results and their understanding during the reading would be essentially higher; This gap can be easily filled, considering the statement "we had the English version of the instrument...";

Thank you very much for your fruitful comments.

Improved

We already put the list of measures in Appendix A.

(2) the representativeness of the sample of "325 usable data" should be justified by authors using the data about overal number of employees of the chosen kind of activity. Now we have enough information about demography of the sample and methods of data collections. However, it is doubtful that the results are reliable in the light of representativeness requirements. I believe that the evidences of sample representativeness will not take a lot of time considering the advanced method of analysis used in the research.

Thank you very much for your fruitful comments.

Improved

We already put some details with regard to the number of employees. Please refer to lines 280-287

Based on the information provided by the company, the total number of salespersons was 683 in 2020. We contacted the company’s VPs and they provided a list of 450 employees. We contacted 450 employees and we got 340 responses. At last, we got 325 usable data out of 340 responses.

We employed Slovin’s formula, we got 284.30 or 284 for the minimum required sample with 983 numbers of the population in 5% margin of error. We believe that 325 respondents are reliable in the light of representativeness.

Reviewer 4 Report

The article follows the scientific structure. The authors need to change or improve the next questions:

-Theoretical framework: the authors can include more updated citations.

-Methods. It is well applied, but to obtain a whole article, I suggest to complete it with a qualitative source (in-deep interview, Delphi…)

-Results: it will be improved with the methodological new tools.

-Conclusions: too much brief. The authors have to expand it.

Author Response

Response to Reviewer 4

This is an interesting study,

Thank you very much for your constructive and kind encouragement. Dear reviewer, we have gone through your constructive suggestions and highlighted our changes in the revised manuscript. Thank you in advance for your time and collaboration to make this manuscript publishable with a high standard and quality.

However, we would like to draw attention to several aspects of the current manuscript that might be improved or needs to be clarified.

Comments

Responses

Theoretical framework: the authors can include more updated citations.

Thank you very much for your fruitful comments.

Improved.

We already include more updated citations in theoretical framework, such as

1.      Khan, M. A., et al. (2022). "Social Undermining and Employee Creativity: The Mediating Role of Interpersonal Distrust and Knowledge Hiding." Behavioral Sciences 12(2).

2.      Scuotto, V., et al. (2022). "An alternative way to predict knowledge hiding: The lens of transformational leadership." Journal of Business Research 140: 76-84.

3.      Zhang, Z., et al. (2022). "Mitigating the negative performance effect of project complexity through an informal mechanism: The conditional mediating role of knowledge hiding." International Journal of Project Management.

Methods. It is well applied, but to obtain a whole article, I suggest to complete it with a qualitative source (in-deep interview, Delphi…)

Thank you very much for your fruitful comments.

Reviewer’s suggestion is well received. We are encouraged by the reviewer’s comments. Our next study will involve a combination of qualitative followed by quantitative analysis. We are in process of evaluating the best qualitative method. However, the current scope of this study is to conduct a quantitative analysis.  We use the reviewer’s recommendation as “suggestions for future research.”

Please refer to lines 502-505

Results: it will be improved with the methodological new tools.

Thank you very much for your fruitful comments.

Thanks to your suggestion we combined fornell-larcker criterion and HTMT (table3 and table 4) to assess the discriminant validity of our measurement model. We employed these methods to get comprehensive results for the discriminant validity.

Please refer to line 328-333

Conclusions: too much brief. The authors have to expand it.

Thank you very much for your fruitful comments.

Improved.

We have written and elaborated the conclusion by adding several sentences related to the result of our study.

Please refer to lines 475-480 (highlighted)

The study findings indicate that knowledge hiding is significantly influenced by the level of distrust. In addition, the level of knowledge complexity has a negative effect on knowledge hiding. It implies that the higher the complexity of knowledge, employees tend to conceal their knowledge. Knowledge hiding behavior has a positive effect on both employees and the organization, such as increasing performance. Regarding the conditional effect of task-relatedness, the finding of this study confirms that task-relatedness weakened the relationship of knowledge complexity and knowledge hiding, whilst strengthening the relationship between distrust and knowledge hiding.

Round 2

Reviewer 4 Report

The authors have included the suggestions. The article can go ahead!